# Quantifying Memory Use in Reinforcement Learning with Temporal Range

## Abstract

How much does a trained RL policy actually use its past observations? We propose *Temporal Range*, a model-agnostic metric that treats first-order sensitivities of multiple vector outputs across a temporal window to the input sequence as a temporal influence profile and summarizes it by the magnitude-weighted average lag. Temporal Range is computed via reverse-mode automatic differentiation from the Jacobian blocks $\partial y_s/\partial x_t \in \mathbb{R}^{c \times d}$ averaged over final timesteps $s \in \{t+1, \ldots, T\}$ and is well-characterized in the linear setting by a small set of natural axioms. Across diagnostic and control tasks (POPGym; flicker/occlusion; Copy-$k$) and architectures (MLPs, RNNs, SSMs), Temporal Range (i) remains small in fully observed control, (ii) scales with the task's ground-truth lag in Copy-$k$, and (iii) aligns with the minimum history window required for near-optimal return as confirmed by window ablations. We also report Temporal Range for a compact Long Expressive Memory (LEM) policy trained on the task, using it as a proxy readout of task-level memory. Our axiomatic treatment draws on recent work on range measures, specialized here to temporal lag and extended to vector-valued outputs in the RL setting. Temporal Range thus offers a practical per-sequence readout of memory dependence for comparing agents and environments and for selecting the shortest sufficient context.

## 1 Introduction

Reinforcement learning (RL) has a long-standing history of utilizing memory to improve performance in complex environments (Hausknecht & Stone, 2015; Berner et al., 2019; Chen et al., 2021; Lu et al., 2023). Examples of machine learning models that incorporate memory include classical Recurrent Neural Networks (RNNs) such as Long Short-Term Memory (LSTM) models (Hochreiter & Schmidhuber, 1997), Transformer (Vaswani et al., 2017), and recently State-Space Models (SSMs) (Gu et al., 2021; Gu & Dao, 2023). However, a rigorous analysis and quantitative measure of the extent to which a *trained* policy utilizes historical information remains largely absent. This matters in partially observed settings: if effective history dependence is short, simpler architectures or shorter contexts suffice; if it is long, we should see it directly in the learned policy rather than infer it from task design or model choice. Current practice relies on indirect signals, such as model class comparisons, environment-specific probes, or sample-complexity bounds (Williams, 2009; Efroni et al., 2022; Morad et al., 2023). These conflate optimization, inductive bias, and true memory demand, and they do not yield a comparable, sequence-level number.

To address this, we formalize a *Temporal Range* metric that aggregates vector-output Jacobians over lags into a magnitude-weighted average look-back, axiomatized for uniqueness. Concretely, for each position $t$ we average Jacobian norms $\|J_{s,t}\|_{\mathrm{mat}}$ over all subsequent final timesteps $s \in \{t+1, \ldots, T\}$, forming per-step weights $w_t = \frac{1}{T-t} \sum_{s=t+1}^{T} \|J_{s,t}\|_{\mathrm{mat}}$, and report

$$\hat{\rho}_T = \frac{\sum_{t=1}^{T} w_t \, (T-t)}{\sum_{t=1}^{T} w_t} \in [0, T-1], \qquad \text{with} \sum_{t=1}^{T} w_t > 0.$$

Thus $\hat{\rho}_T$ answers "how far back is this policy looking *here*?" at the level of a specific rollout and timestep. Our theoretical starting point is the axioms of range from Bamberger et al. (2025); we specialize them to temporal lag, extend to vector-valued outputs, and study their consequences for reinforcement learning agents.

For vector-output *linear* maps we give a short axiomatic derivation that fixes both the unnormalized and normalized forms: single-step calibration, additivity over disjoint time indices (or magnitude-weighted averaging), and absolute homogeneity identify the unique matrix-norm–weighted lag sums/averages. The same formulas applied to the local linearization yield our nonlinear policy metric. The normalized form is invariant to uniform input rescaling (change of units) and uniform output rescaling, making cross-agent and cross-environment comparisons straightforward.

Computationally, Temporal Range is inexpensive. The required Jacobian blocks are *policy* derivatives with respect to observation inputs and can be obtained with standard reverse-mode automatic differentiation on the policy alone. When direct auto-differentiation on a given model is impractical, we train a compact LEM policy on the same task and compute the same quantities on this proxy.

We validate Temporal Range across POPGym diagnostics and classic control with Multi-Layer Perceptrons (MLPs), gated RNNs (e.g., LSTMs, Gated Recurrent Units (GRUs) (Chung et al., 2014), and LEM), as well as SSMs. The metric (i) stays near zero in fully observed control, (ii) scales with the ground-truth offset in Copy-$k$, and (iii) aligns with the smallest history window needed to achieve near-optimal return, as verified by window ablations that rebuild hidden state from truncated histories. These results make Temporal Range a practical tool for auditing memory use and for selecting the shortest sufficient context.

**Contributions.** (i) We formalize a simple, model-agnostic metric, Temporal Range, that aggregates per-timestep *matrix*-norm Jacobians of a *vector* output into a magnitude-weighted average lag, computable in one reverse-mode pass. This captures temporal dependence and is complementary to existing saliency and perturbation-style explanations, which focus on spatial attention within single frames. (ii) A minimal axiomatic justification for vector-output linear maps fixing both unnormalized and normalized forms, with invariance to uniform input and output rescaling. (iii) Empirical validation on diagnostics and control showing agreement with ground truth and with window-size requirements for high return. (iv) A LEM-based proxy enabling use when the policy blocks gradients or is black-box.

## 2 RELATED WORK

Research on memory in RL has often focused on *implementing* mechanisms (RNNs, Transformers, memory buffers), but fewer works directly *measure* how much past information agents actually use. We review theoretical foundations and empirical approaches to quantifying memory dependence.

**Foundational context.** Learning long-range dependencies is nontrivial due to exploding/vanishing gradients and bias from truncated BPTT (Pascanu et al., 2013; Tallec & Ollivier, 2018). Our focus on history use is motivated by the classical view that partial observability demands memory (Kaelbling et al., 1998) and by temporal credit-assignment foundations in RL (Sutton & Barto, 2018). Computationally, our reverse-mode computation of temporal influence ties directly to backpropagation through time (Werbos, 1990). Empirically, our scalar Temporal Range parallels "effective context length" observations in language modeling (Khandelwal et al., 2018). Finally, since our metric aggregates Jacobian magnitudes, we note standard cautions from the saliency literature (Adebayo et al., 2018).

### 2.1 THEORETICAL FRAMEWORKS

Classical Markov decision processes (MDPs) assume the Markov property: future states depend only on the present. Partially observed Markov decision processes (POMDPs), however, are non-Markovian to the agent and thus require memory. Mizutani & Dreyfus (2017) show that recurrent architectures can render non-Markovian processes Markovian in an augmented state space. Efroni et al. (2022) prove sample-complexity bounds for POMDPs where latent states can be decoded from histories of length $m$, showing exponential scaling in $m$. Other work links partial observability and memory through Bayesian/active inference perspectives (Malekzadeh & Plataniotis, 2024), and extends to decentralized multi-agent settings (Omidshafiei et al., 2017).

## 2.2 EMPIRICAL MEASUREMENT APPROACHES

**Model comparisons.** POPGym (Morad et al., 2023) provides partially observable tasks for comparing memory-augmented models.

**Environment-specific studies.** Modified CartPole (Koffi et al., 2020), Pacman (Fallas-Moya et al., 2021), and multi-robot delivery (Omidshafiei et al., 2017) isolate memory requirements in specific tasks.

**Performance gaps.** Studies such as Meng et al. (2021) compare memoryless Twin Delayed Deep Deterministic Policy Gradient (TD3) vs. recurrent (LSTM-TD3) agents in MDPs vs. POMDPs, quantifying degradation without memory. Williams (2009) give conditions where stochastic memoryless policies suffice.

**Robustness and complexity.** Metrics such as adversarial perturbation robustness (Zhang et al., 2020) and theoretical sample-complexity bounds (Efroni et al., 2022) indirectly quantify the cost of memory.

**Gradient-based attribution and deep RL.** Our metric builds on the classical saliency view that interprets Jacobian magnitudes, $|\partial y/\partial x|$, as input importance (Simonyan et al., 2014). In deep RL, prior work uses such saliency/perturbation maps to show *where* a vision policy attends within a single frame (Greydanus et al., 2018), but these analyses are inherently *spatial* and per-timestep and often note that raw Jacobian maps can be visually uninformative. Gradient-based attributions admit known caveats; complementary baselines include Integrated Gradients, SmoothGrad, and influence-function–style analyses (Sundararajan et al., 2017; Smilkov et al., 2017; Koh & Liang, 2020; Pruthi et al., 2020).

**Temporal Range versus saliency.** By contrast, Temporal Range is explicitly *temporal*: we aggregate matrix-norm Jacobian blocks over past observations to produce a scalar that quantifies *how far back* a policy depends on history. Beyond attribution, we supply (i) an axiomatic justification that fixes the form of the metric and (ii) behavioral validation via window ablations aligning measured range with the minimal history needed for high return. To our knowledge, no prior saliency-based method offers a scalar, sequence-level readout of memory dependence with these guarantees (Simonyan et al., 2014; Greydanus et al., 2018).

## 2.3 BENCHMARK DEVELOPMENT

Benchmarks provide standardized evaluation. POPGym (Morad et al., 2023) is widely used, while newer efforts like MIKASA (Cherepanov et al., 2025) classify memory-intensive RL tasks such as robotic manipulation.

## 3 A TEMPORAL-RANGE MEASURE FOR REINFORCEMENT LEARNING

How much does a policy at time $T$ look back in the past? Our goal is a concrete, *model-agnostic* number. The plan: treat first-order sensitivities of the final *vector* output with respect to earlier inputs as a *temporal influence profile*, and summarize that profile by the expected look-back (average lag). We then show this summary is not arbitrary: in the linear case, it is the *unique* object that satisfies a small set of first-principles requirements for any reasonable "how-far-back" score. For nonlinear policies, we apply the same formula to the local linearization.

### 3.1 SETUP AND DEFINITION

Let $F : \mathbb{R}^{T \times d} \to \mathbb{R}^c$ denote the map from a length-$T$ observation sequence to a vector output, $X_{1:T} \mapsto y(X)$, where $X_{1:T} = [x_1, \ldots, x_T]$ with $x_t \in \mathbb{R}^d$ and $y(X) \in \mathbb{R}^c$ is the policy's vector output (e.g., action logits or probabilities). For each pair $s, t$ with $1 \leq t < s \leq T$, write the Jacobian block

$$J_{s,t}(X) := \frac{\partial y_s(X)}{\partial x_t} \in \mathbb{R}^{c \times d},$$

where $y_s(X)$ is the output at step $s$ given the partial sequence $X_{1:s}$. Fix a matrix norm $\|\cdot\|_{\mathrm{mat}}$ on $\mathbb{R}^{c\times d}$ (e.g., Frobenius, or an induced operator norm). For a window of length $T$, define the per-step *influence weight* at position $t$ by aggregating the Jacobian norms over all subsequent timesteps $s \in \{t+1, \ldots, T\}$ via an operator $\phi$:

$$w_t(X) := \phi_{s=t+1}^T \|J_{s,t}(X)\|_{\mathrm{mat}} \quad \geq 0. \tag{1}$$

The operator $\phi$ is configurable: we use $\phi = \mathrm{mean}$ (i.e., $\frac{1}{T-t}\sum_{s=t+1}^T$) throughout this paper, as it provides better discrimination between tasks with different memory requirements. For tasks with concentrated temporal dependencies (e.g., a single critical observation affecting one future action), $\phi = \max$ may be more appropriate; see Appendix A.6 for a comparison. In practice, we compute $w_t$ by averaging over a calibration set of multiple rollouts to reduce variance. The *lag* is defined by

$$\ell(t) := T - t \in \{1, \ldots, T-1\}. \tag{2}$$

The map $t \mapsto w_t(X)$ can be interpreted as a nonnegative temporal influence profile; we summarize it by the expected lag under these weights.

Note that throughout, we assume the vector outputs across the window are not independent of the input sequence $X_{1:T}$ at the evaluation point $X$, i.e., at least one Jacobian block $J_{s,t}(X)$ is nonzero for some $s, t$. Equivalently, $\sum_{t=1}^T w_t(X) > 0$.

**Definition 3.1** (Temporal range). For differentiable $F$ and an evaluation sequence $X$,

$$\rho_T(F; X) := \sum_{t=1}^T w_t(X)\,\ell(t), \tag{3}$$

$$\hat{\rho}_T(F; X) := \frac{\sum_{t=1}^T w_t(X)\,\ell(t)}{\sum_{t=1}^T w_t(X)}. \tag{4}$$

It follows directly that $0 \leq \hat{\rho}_T(F; X) \leq T-1$. The unnormalized $\rho_T$ aggregates both *how far back* and *how strongly* the past matters; the normalized $\hat{\rho}_T$ reports an average look-back in steps.

## 3.2 AXIOMATIC BASIS

We introduce *Temporal Range* and give a short axiomatic justification tailored to time and vector outputs. Our axioms follow the style of prior "range" functionals but are adapted to temporal lag $\ell(t) = T-t$ and matrix norms on vector outputs, yielding in the *linear* case $L(z_1, \ldots, z_T) = \sum_{t=1}^T B_t z_t$ with $B_t \in \mathbb{R}^{c\times d}$ the unique forms:

$$\rho_T(L) = \sum_{t=1}^T \|B_t\|_{\mathrm{mat}}\,(T-t), \qquad \hat{\rho}_T(L) = \frac{\sum_{t=1}^T \|B_t\|_{\mathrm{mat}}\,(T-t)}{\sum_{t=1}^T \|B_t\|_{\mathrm{mat}}}. \tag{5}$$

We then use these linear forms at the local linearization of a nonlinear policy. Full axioms (our temporal/vector-output variant) and uniqueness proofs appear in App. A.3. See Bamberger et al. (2025) for related axiomatization style.

## 3.3 FROM LINEAR MAPS TO POLICIES

Given a differentiable policy that produces vector outputs $y_s(X) \in \mathbb{R}^c$ at each step $s$ in a window of length $T$, define the Jacobian blocks and their averaged matrix–norm magnitudes

$$J_{s,t}(X) = \frac{\partial y_s}{\partial x_t}(X) \in \mathbb{R}^{c\times d}, \qquad w_t(X) = \frac{1}{T-t}\sum_{s=t+1}^T \|J_{s,t}(X)\|_{\mathrm{mat}}. \tag{6}$$

Plugging these weights into (5) yields

$$\rho_T(F; X) = \sum_{t=1}^T w_t(X)\,(T-t),$$

$$\hat{\rho}_T(F; X) = \frac{\sum_{t=1}^T w_t(X)\,(T-t)}{\sum_{t=1}^T w_t(X)}, \tag{7}$$

the magnitude-weighted average look-back of the local linearization at $X$.

## 3.4 INVARIANCES AND SCOPE

**Uniform output rescaling.** If $\tilde{y}_s = \alpha\, y_s$ for all $s$ with $\alpha \neq 0$, then $J_{s,t}(\tilde{y}) = \alpha\, J_{s,t}(y)$ for all $s, t$, so every term in the averaging sum $w_t = \frac{1}{T-t}\sum_{s=t+1}^{T} \|J_{s,t}\|_{\mathrm{mat}}$ is multiplied by $|\alpha|$, leaving $\hat{\rho}_T$ unchanged (while $\rho_T$ rescales by $|\alpha|$). This covers, e.g., temperature scaling of logits.

**Uniform input rescaling (change of units).** If we change units by $x_t^\star = \beta x_t$ with $\beta \neq 0$, then by the chain rule $\|\frac{\partial y_s}{\partial x_t^\star}(X)\|_{\mathrm{mat}} = \frac{1}{|\beta|}\|\frac{\partial y_s}{\partial x_t}(X)\|_{\mathrm{mat}}$ for all $s, t$, so every term in $w_t$ is multiplied by $1/|\beta|$, leaving $\hat{\rho}_T$ invariant (while $\rho_T$ rescales by $1/|\beta|$). This concerns a *reparameterization of inputs*; feeding $\beta X$ into a fixed network changes $F$ and may change $\hat{\rho}_T$.

## 3.5 IMMEDIATE RL CONSEQUENCES

Temporal Range serves primarily as an interpretability tool: it quantifies how far back a trained policy looks when making decisions. In partially observed or noisy settings, earlier inputs matter for state reconstruction and smoothing, so $\hat{\rho}_T$ grows. Comparing architectures, short ranges on tasks believed to need memory may indicate under-capacity; long ranges on near-Markov tasks indicate unnecessary slow modes (common with some SSMs). The measure also provides an upper bound for sizing history windows: aggregating $\hat{\rho}_T$ across evaluation episodes suggests a conservative context length, which window ablations can then validate.

Temporal Range is a local, per-sequence diagnostic: at a given rollout window it asks how far back the policy is effectively looking. In fully observed control, if each $y_s$ depends only on $x_s$, then $J_{s,t}(X) = 0$ for all $t < s$, so $w_t = 0$ and $\hat{\rho}_T = 0$ at that point. With a finite effective memory $(x_{s-m+1:s})$, $J_{s,t} = 0$ for $t \leq s - m$ across all $s$ and $\hat{\rho}_T \in [0, m-1]$; for a single-offset dependence (Copy-$k$), $\hat{\rho}_T = k$ exactly. These cases calibrate scale and serve as unit tests.

Policy and value heads can differ materially. Applying the same computation to the value head $V_T$ often yields a larger effective range, reflecting return propagation; a gap between policy and value ranges helps explain unstable advantage estimates and motivates retuning temporal credit assignment (e.g., the $\lambda$ in Generalized Advantage Estimation (GAE) (Schulman et al., 2016)). Importantly, uniform output rescaling (e.g., logits temperature) multiplies all $w_t$ by the same factor and leaves the normalized range invariant, so exploration via temperature does not confound comparisons of $\hat{\rho}_T$.

## 3.6 ANALYTICAL CALIBRATION IN TOY DIFFERENTIABLE SETTINGS

When the end-to-end map happens to be fully differentiable and structured, Temporal Range admits closed-form expressions that calibrate the measure. In App. A.4 we work out two cases: (*i*) COPY-$k$, where $\hat{\rho}_T = k$ exactly for any matrix norm, and (*ii*) linear recurrent readouts, where $w_t = \|QA^{T-t}C\|_{\mathrm{mat}}$ yields a profile governed by the spectrum of $A$. These derivations are useful for calibration and intuition. In practical RL environments, we compute Jacobians via reverse-mode *on the policy*. If the target model is not amenable to automatic differentiation, we compute the same quantities on a compact LEM proxy (Sec. 4).

## 4 APPROXIMATING TEMPORAL RANGE WITH LEM

Computing Temporal Range requires differentiating the policy outputs $y_s$ for $s \in \{1, \ldots, T\}$ with respect to past *observations* $x_{1:T}$. This does not require differentiating the simulator. In settings where the policy itself is non-differentiable or inaccessible to autograd, we train a compact LEM policy on the same task and compute TR on this proxy. We use Long Expressive Memory (LEM) (Rusch et al., 2022) as an effective proxy thanks to its stable long-horizon gradients. LEM is designed for long-horizon sequence modeling and maintains stable gradients over extended contexts via a multiscale ordinary differential equation (ODE) formulation.

**Empirical check.** In benchmarks, proxy $\hat{\rho}_T$ varies in line with known task structure: it increases with $k$ in Copy-$k$ and grows under partial observability, indicating that the proxy yields reliable Jacobians when the original model is not amenable to differentiation.

## 5 EXPERIMENTS

We quantify how much trained agents use history by measuring *Temporal Range* $\hat{\rho}_T$ and validating it with *window ablations*. We evaluate four architectures, namely LEM, GRU, LSTM, and Linear Oscillatory State-Space models (LinOSS) (Rusch & Rus, 2025), across diagnostic and control settings (Tables 2–3).

### 5.1 SETUPS

**Training.** All policies are trained with Proximal Policy Optimization (PPO) ((Schulman et al., 2017)) for $10^7$ steps. Unless stated, the actor–critic uses a single LEM cell (size 128) with 64-d encoder/decoder MLPs; GRU/LSTM and LinOSS are dimension-matched. We use the JAX auto-differentiation framework (Bradbury et al., 2018).

**Temporal-Range computation.** At evaluation, for each rollout of length $T$ we compute Jacobian blocks $J_{s,t} = \partial y_s / \partial x_t \in \mathbb{R}^{c \times d}$ for all pairs $s, t$ with $t < s \leq T$, where $y_s$ is the *action-vector* output at step $s$. We then average these norms over final timesteps to form per-step weights $w_t = \frac{1}{T-t} \sum_{s=t+1}^{T} \|J_{s,t}\|_{\mathrm{mat}}$ and aggregate into $\hat{\rho}_T$ (Def. 3.1). Unless otherwise noted, we use the Frobenius norm. We further average over multiple rollouts (calibration set) and report means and standard deviations over episodes.

**Window ablations (behavioral validation).** To test whether measured look-back is functionally required, we evaluate each trained policy under truncated histories of size $m \in \{1, 2, 4, 8, 16, 32, 64\}$: at every step, we rebuild hidden state from the last $m$ observations only. Curves (return vs. $m$) and the summary (Best@m, Avg.) are given in Fig. 1 and Table 4.

### 5.2 ENVIRONMENTS

We evaluate agents on a small suite of POPGym ((Morad et al., 2023)) environments designed to probe distinct memory behaviors:

- **Control (fully vs. partially observed).** CartPole is nearly Markov and should require little history. In *Stateless CartPole*, positions are hidden, so the agent must integrate observations. The *Noisy Stateless* variant adds observation noise, increasing smoothing demand.
- **Diagnostics (known lags).** *Repeat First* asks the agent to recall an early value later in the episode. *Copy-k* (with $k \in \{1, 3, 5, 10\}$) requires outputting the observation from exactly $k$ steps ago, providing ground-truth offsets.

## 6 RESULTS

We now present empirical results. Tables report summary statistics across environments and architectures, while plots pair reward traces with temporal influence profiles to visualize both performance and measured look-back. Unless otherwise noted, results use LEM as the representative model; the appendix contains full sweeps and additional figures.

### 6.1 SUMMARY ACROSS SETTINGS

Figure 1 (ablations) and Figure 2 (profiles) show the same tasks side by side. On *CartPole* (near-Markov), effective look-back is short (GRU $\sim$1, LEM $\sim$3; Table 2), and returns saturate at small windows, matching the compact profiles. In *Stateless CartPole*, profiles develop longer tails and ablations recover only with larger windows. In *Copy-k*, profiles shift right with $k$ and ablations exhibit knees near $\hat{\rho}_T + 1$, while calibration MAE (Table 5) is lowest for GRU at higher $k$, with LEM/LinOSS tending to overestimate due to slow/multiscale modes.

### 6.2 KEY OBSERVATIONS

Range tracks observability: it is small in *CartPole* and increases in *Stateless CartPole*, with ablations recovering only once windows exceed the measured look-back (Figs. 1, 2; Tables 2, 4). In *Copy-*

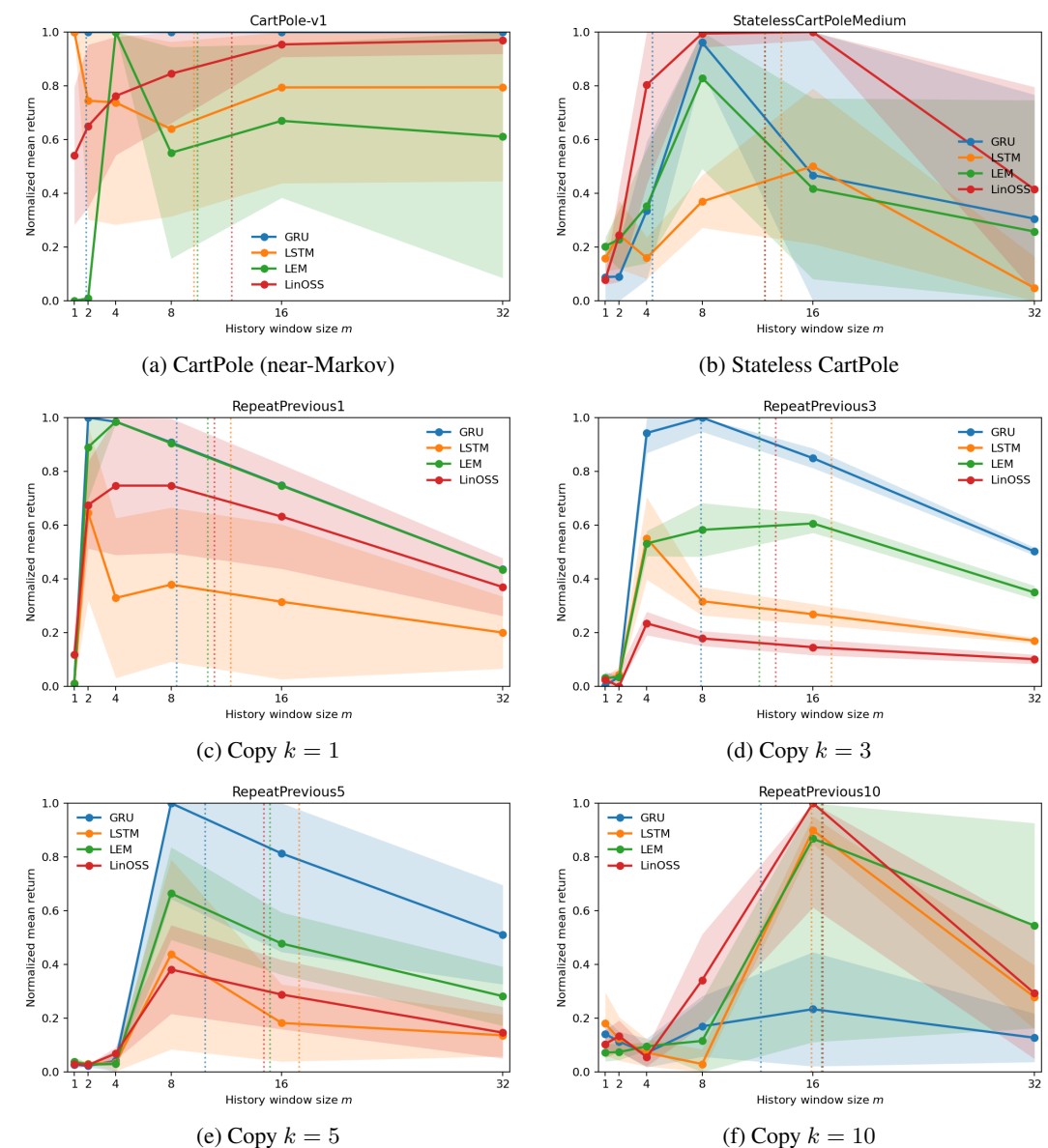

Figure 1: **Window ablations.** Normalized return vs. context window size $m$ across architectures. Dotted vertical lines show $\hat{\rho}_T$ values. Performance recovers when $m$ exceeds temporal range, confirming TR identifies minimum sufficient context. Note that $\hat{\rho}_T$ often aligns remarkably well with task requirements (e.g., GRU's $\hat{\rho}_T \approx 12$ for Copy $k = 10$). When TR appears to fall short of the empirical peak (e.g., $\hat{\rho}_T = 12$ while peak occurs at $m = 16$), this is typically an artifact of our sparse window sampling ($m \in \{1, 2, 4, 8, 16, 32\}$); the true performance peak likely lies between tested values, closer to the TR prediction.

$k$, $\hat{\rho}_T$ scales with $k$ and the ablation knee aligns with ground truth (Tables 2, 5). Notably, even in near-Markovian CartPole, restricting context to only the current observation ($m = 1$) causes performance degradation, likely due to distribution shift; models trained with full history adapt to using second-to-last observations, making sudden truncation disruptive. Interestingly, TR reveals architecture-specific inefficiencies: GRU achieves $\hat{\rho}_T \approx 2$ on CartPole, while LEM and LSTM hover around 10, indicating they genuinely rely on longer history despite the task's near-Markovian nature. This highlights TR's value as an interpretability tool for exposing how different architectures use memory, even when such usage is unnecessary for performance. Across architectures, GRU tends

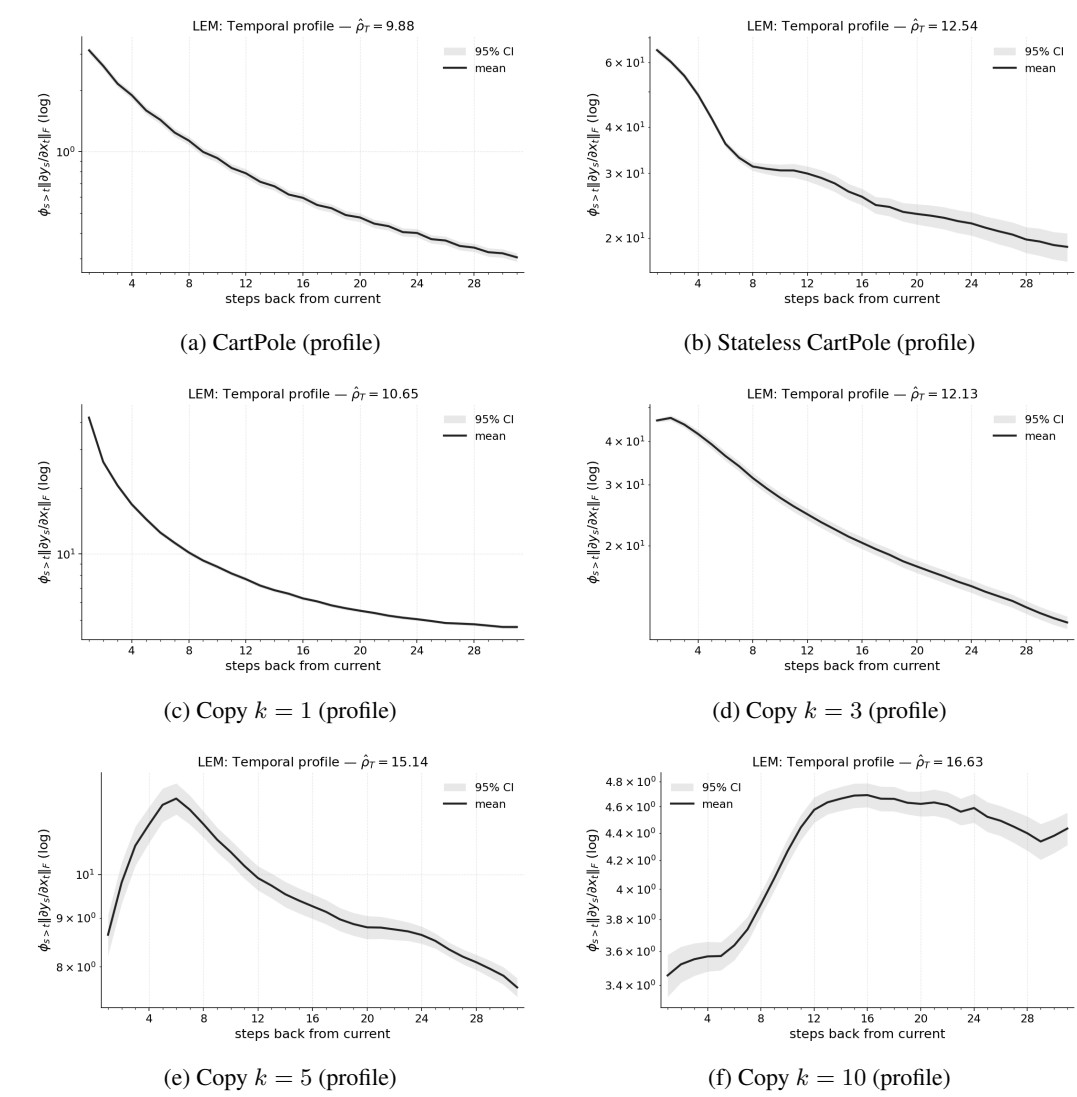

**Figure 2: Temporal influence profiles (LEM).** Jacobian magnitude vs. steps back from current timestep. Profiles show how policy depends on observation history: CartPole concentrates on recent steps, Stateless CartPole distributes broadly, Copy-$k$ peaks near required lookback distance.

to shorter effective memory unless forced; LEM maintains multiscale tails with stable behavior; LinOSS often inflates range via slow modes on easy tasks; LSTM is more seed-sensitive.

## 6.3 TR-GUIDED MEMORY-EFFICIENT DEPLOYMENT

To validate TR's practical utility for architecture design, we test whether TR-recommended context windows enable memory-efficient deployment while preserving performance. We follow a two-stage protocol: (1) train small LEM models (hidden size 128) to compute TR estimates $\hat{\rho}_T$, (2) train large LEM models (hidden size 512) with full continuous hidden state evolution. We then evaluate the large models under three regimes: *training* (continuous evolution baseline), *TR window* (hidden states rebuilt from observation buffers of length $\lceil \hat{\rho}_T+1 \rceil$), and *half-TR* (buffers of length $\lceil (\hat{\rho}_T+1)/2 \rceil$). This tests whether windowed evaluation approximates the model's natural operation while reducing memory requirements.

Performance is normalized to $[0, 1]$ per environment using global min/max values from all training runs, with results averaged over multiple evaluation trials.

Table 1: TR-guided window validation: normalized performance [0,1] (mean $\pm$ std over evaluation trials). Training baseline shows final performance with continuous hidden state evolution. Retention indicates windowed performance relative to training baseline.

| Environment | TR | Training | TR window | Half-TR | Retention |
|---|---|---|---|---|---|
| Noisy Stateless CartPole | 12.3 | $0.474 \pm 0.017$ | $0.953 \pm 0.068$ | $0.637 \pm 0.116$ | 201.1% |
| Copy $k=3$ | 12.1 | $0.934 \pm 0.007$ | $0.970 \pm 0.038$ | $0.838 \pm 0.021$ | 103.9% |
| Copy $k=10$ | 16.6 | $0.561 \pm 0.018$ | $0.979 \pm 0.034$ | $0.052 \pm 0.027$ | 174.1% |

Table 2: Temporal Range $\hat{\rho}_T$ (mean $\pm$ std steps). Values increase with task memory requirements where policies are competent: moderate for near-Markovian CartPole (with architecture-specific variation), higher for stateless variants, and scaling with $k$ in Copy tasks.

| Environment | LEM | GRU | LSTM | LinOSS |
|---|---|---|---|---|
| CartPole | $9.883 \pm 0.358$ | $1.854 \pm 1.842$ | $9.642 \pm 0.818$ | $12.362 \pm 0.377$ |
| Stateless CartPole | $12.536 \pm 1.428$ | $4.391 \pm 1.000$ | $13.704 \pm 2.048$ | $12.514 \pm 1.101$ |
| Noisy Stateless CartPole | $12.261 \pm 0.637$ | $9.762 \pm 1.388$ | $15.274 \pm 1.342$ | $14.151 \pm 0.488$ |
| RepeatFirst | $17.793 \pm 2.536$ | $3.742 \pm 2.504$ | $12.489 \pm 2.174$ | $21.177 \pm 0.813$ |
| Copy $k = 1$ | $10.647 \pm 0.424$ | $8.398 \pm 1.552$ | $12.294 \pm 0.802$ | $11.111 \pm 0.296$ |
| Copy $k = 3$ | $12.126 \pm 0.513$ | $7.916 \pm 0.498$ | $17.312 \pm 0.880$ | $13.298 \pm 0.956$ |
| Copy $k = 5$ | $15.137 \pm 0.588$ | $10.453 \pm 1.538$ | $17.255 \pm 1.176$ | $14.693 \pm 0.670$ |
| Copy $k = 10$ | $16.625 \pm 0.585$ | $12.224 \pm 1.376$ | $15.900 \pm 3.403$ | $16.715 \pm 0.652$ |

The results demonstrate that TR-recommended windows successfully approximate training-regime performance despite the distribution shift from periodic hidden state reconstruction. Models evaluated with TR-guided truncated windows maintain high retention relative to continuous evolution (103.9–201.1%), validating that TR identifies sufficient context for the windowed regime. In contrast, half-TR windows show substantial degradation, confirming that truncating below TR recommendations loses critical temporal information. These findings support the hypothesis that TR computed from small models (h=128) provides actionable upper bounds for deploying large models (h=512) with reduced memory. The ability to maintain performance while reconstructing hidden states from windows of 10–20 steps (versus continuous evolution over full episodes) demonstrates TR's utility for memory-constrained deployment scenarios.

## 7 DISCUSSION

**What range captures.** Temporal Range summarizes *local, first-order* influence of the input history on the final vector output via matrix-norm Jacobians. It answers "how far back is the policy looking *here*?" at a specific rollout point. In our paired views, when the temporal profile concentrates near small lags (Fig. 2a), returns saturate with short windows (Fig. 1a); when profiles carry longer tails (e.g., Fig. 2b), performance recovers only once the window exceeds the measured look-back (Fig. 1b; Table 4).

**Reading range with reward.** Range is not reward. On near-Markov control, long tails (e.g., slow modes) can be unnecessary yet harmless; under partial observability or noise, larger ranges are often necessary but not sufficient. The alignment between ablations and profiles (Figs. 1, 2) and the aggregates (Tables 2, 3, 4) provides a consistent behavioral cross-check: short profiles go with early saturation, while longer-tailed profiles demand larger windows before returns improve.

**Practical guidance.** Use $\hat{\rho}_T$ to audit memory use and choose the *shortest sufficient* context: (i) if return is high and $\hat{\rho}_T$ is large on a near-Markov task, simplify the architecture or shorten context; (ii) if return is capped and $\hat{\rho}_T$ is small on a partially observed task, increase memory capacity or training horizon; (iii) if noise increases $\hat{\rho}_T$ without return gains, you are smoothing more without extracting signal—revisit representation or denoising.

Table 3: Policy performance (normalized return in [0,1], mean $\pm$ std). After hyperparameter tuning, all architectures achieve strong performance, enabling meaningful TR analysis across models.

| Environment | LEM | GRU | LSTM | LinOSS |
|---|---|---|---|---|
| CartPole | $0.989 \pm 0.002$ | $0.998 \pm 0.003$ | $0.960 \pm 0.029$ | $0.914 \pm 0.135$ |
| Stateless CartPole | $0.892 \pm 0.186$ | $0.999 \pm 0.002$ | $0.926 \pm 0.065$ | $0.796 \pm 0.320$ |
| Noisy Stateless CartPole | $0.534 \pm 0.010$ | $0.492 \pm 0.027$ | $0.408 \pm 0.022$ | $0.469 \pm 0.063$ |
| RepeatFirst | $0.541 \pm 0.033$ | $0.356 \pm 0.132$ | $1.000 \pm 0.000$ | $0.912 \pm 0.077$ |
| Copy $k = 1$ | $0.975 \pm 0.014$ | $0.983 \pm 0.019$ | $0.435 \pm 0.069$ | $0.850 \pm 0.251$ |
| Copy $k = 3$ | $0.824 \pm 0.055$ | $0.949 \pm 0.032$ | $0.394 \pm 0.031$ | $0.455 \pm 0.051$ |
| Copy $k = 5$ | $0.455 \pm 0.033$ | $0.647 \pm 0.124$ | $0.325 \pm 0.013$ | $0.389 \pm 0.012$ |
| Copy $k = 10$ | $0.517 \pm 0.045$ | $0.465 \pm 0.009$ | $0.407 \pm 0.035$ | $0.511 \pm 0.023$ |

## 8 CONCLUSION

We presented *Temporal Range*, a first-order, model-agnostic measure of how far back a trained policy effectively looks. By turning vector-output Jacobian blocks into a temporal influence profile and summarizing by a magnitude-weighted average lag, Temporal Range provides a single, interpretable number per sequence and timestep. An axiomatic derivation for vector-output linear maps fixes the form of both unnormalized and normalized variants and yields invariance to uniform input and output rescaling. The metric is practical, taking one reverse-mode pass per sequence, and, when the policy is non-differentiable or black-box, a proxy LEM policy supplies reliable Jacobians.

Across POPGym diagnostics and control, Temporal Range is small in fully observed control, scales with fixed task lags (Copy-$k$), and matches the minimum history window required for near-optimal return as confirmed by window ablations. This makes it useful for auditing memory dependence, comparing agents and environments, and choosing the shortest sufficient context.

Limitations include locality (the measure is specific to the evaluated rollout/time), dependence on preprocessing and norm choice, and the possibility that slow modes inflate range without improving return. Future work includes time-resolved profiles across decisions, per-output-component ranges (policy vs. value), causal perturbation checks, and regularizers that penalize unnecessary range to bias training toward simpler, shorter-context solutions.

### REPRODUCIBILITY STATEMENT

All code to reproduce our tables, figures, and ablations is available in an anonymous repository: `https://anonymous.4open.science/r/TemporalRange-26E4/README.md`. The repo includes end-to-end training/evaluation scripts for PPO agents, Jacobian/Temporal Range computation utilities, and the window-ablation driver. Unless otherwise noted in the text, we use the hyperparameters listed in Table 7. We provide fixed random seeds and configuration files to regenerate the reported runs (curves use multiple seeds as indicated in captions; summary tables average across episodes/trials as stated). The repository also includes plotting code and exact evaluation commands to reproduce Tables 2–4 and Figs. 1–5 from raw rollouts.

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

# A APPENDIX

## A.1 WINDOW ABLATION RESULTS

Table 4: Window ablation summary (normalized to [0,1]). Best@m shows peak performance and required window size; Avg. shows mean across all windows.

| Environment | LEM | GRU | LSTM | LinOSS |
|---|---|---|---|---|
| CartPole | 1.000@4 / 0.449 | 1.000@1 / 1.000 | 1.000@1 / 0.744 | 1.000@64 / 0.817 |
| Stateless CartPole | 0.829@8 / 0.327 | 0.961@8 / 0.335 | 0.501@16 / 0.215 | 1.000@16 / 0.512 |
| Noisy Stateless CartPole | 0.809@16 / 0.506 | 1.000@16 / 0.574 | 0.767@16 / 0.471 | 0.655@8 / 0.314 |
| RepeatFirst | 1.000@1 / 0.396 | 0.682@8 / 0.328 | 0.719@16 / 0.525 | 0.944@16 / 0.512 |
| Copy $k = 1$ | 0.985@4 / 0.567 | 1.000@2 / 0.585 | 0.645@2 / 0.270 | 0.747@4 / 0.473 |
| Copy $k = 3$ | 0.606@16 / 0.309 | 1.000@8 / 0.480 | 0.551@4 / 0.201 | 0.234@4 / 0.101 |
| Copy $k = 5$ | 0.664@8 / 0.217 | 1.000@8 / 0.348 | 0.437@8 / 0.126 | 0.381@8 / 0.136 |
| Copy $k = 10$ | 0.868@16 / 0.257 | 0.233@16 / 0.136 | 0.899@16 / 0.226 | 1.000@16 / 0.276 |

## A.2 COPY-$k$ CALIBRATION

Table 5: Copy-$k$ calibration: MAE($\hat{\rho}_T$, $k$) over episodes. Lower values indicate better alignment between measured TR and ground-truth memory requirement $k$.

| Environment | LEM | GRU | LSTM | LinOSS |
|---|---|---|---|---|
| Copy $k = 1$ | $9.65 \pm 0.42$ | $7.40 \pm 1.55$ | $11.29 \pm 0.80$ | $10.11 \pm 0.30$ |
| Copy $k = 3$ | $9.13 \pm 0.51$ | $4.92 \pm 0.50$ | $14.31 \pm 0.88$ | $10.30 \pm 0.96$ |
| Copy $k = 5$ | $10.14 \pm 0.59$ | $5.45 \pm 1.54$ | $12.25 \pm 1.18$ | $9.69 \pm 0.67$ |
| Copy $k = 10$ | $6.63 \pm 0.58$ | $2.28 \pm 1.28$ | $5.91 \pm 3.38$ | $6.72 \pm 0.65$ |

## A.3 DERIVING TEMPORAL RANGE FROM AXIOMS (VECTOR-OUTPUT LINEAR MAPS)

Inspired by recent axiomatic treatments of "range" in other domains (Bamberger et al., 2025), we show that in the temporal setting a small set of natural conditions uniquely fix both the unnormalized and normalized forms. We first characterize the linear case to fix the form of any reasonable "how-far-back" score with vector outputs and matrix norms. Consider a length-$T$ linear map

$$L(z_1, \ldots, z_T) = \sum_{t=1}^{T} B_t z_t \qquad (B_t \in \mathbb{R}^{c \times d}), \tag{8}$$

where $z_t \in \mathbb{R}^d$ is the input at time $t$, and $\ell(t) = T - t$. We ask for a score $\rho_T(L) \in (0, \infty)$ satisfying:

**R1-u (single-step calibration with magnitude).** If $L(z) = B z_{T-k}$ with any nonzero $B \in \mathbb{R}^{c \times d}$, then $\rho_T(L) = \|B\|_{\mathrm{mat}} k$.

**R2 (additivity over disjoint times).** If $L_1$ and $L_2$ depend on disjoint time indices, then $\rho_T(L_1 + L_2) = \rho_T(L_1) + \rho_T(L_2)$.

**R3 (absolute homogeneity).** For any $\alpha \in \mathbb{R}$, $\rho_T(\alpha L) = |\alpha| \, \rho_T(L)$.

**Proposition A.1** (Uniqueness of the unnormalized form for vector outputs)**.** *Fix any matrix norm* $\|\cdot\|_{mat}$ *on* $\mathbb{R}^{c \times d}$. *There is a unique nonnegative map* $\rho_T$ *on linear maps obeying* R1-u–R3, *namely*

$$\rho_T(L) = \sum_{t=1}^{T} \|B_t\|_{mat} \, \ell(t). \tag{9}$$

*Proof of Proposition A.1.* Let $L_t$ denote the projection $L_t(z) = z_t$. Then $L = \sum_{t=1}^{T} B_t L_t$, and $\{B_t L_t\}$ depend on disjoint time indices. By R2 and R3,

$$\rho_T(L) = \sum_{t=1}^{T} \rho_T(B_t L_t) = \sum_{t=1}^{T} \|B_t\|_{\mathrm{mat}} \, \rho_T\Big(\tfrac{B_t}{\|B_t\|_{\mathrm{mat}}} L_t\Big).$$

By single-step calibration (R1-u), $\rho_T(\frac{B_t}{\|B_t\|_{\mathrm{mat}}} L_t) = \ell(t)$, yielding $\rho_T(L) = \sum_{t=1}^{T} \|B_t\|_{\mathrm{mat}} \ell(t)$. Uniqueness follows because R1-u fixes single-index values and R2–R3 propagate to disjoint sums. □

For an average lag, replace additivity with magnitude-weighted averaging.

**R4 (magnitude-weighted averaging).** If $L_1$ and $L_2$ depend on disjoint time indices and are nonzero, then for any $\alpha, \beta \in \mathbb{R}$ with $(\alpha, \beta) \neq (0,0)$ and $\alpha L_1 + \beta L_2 \neq 0$,

$$\hat{\rho}_T(\alpha L_1 + \beta L_2) = \frac{|\alpha|\, \hat{\rho}_T(L_1) + |\beta|\, \hat{\rho}_T(L_2)}{|\alpha| + |\beta|}.$$

**R1-n (single-step calibration without magnitude).** If $L(z) = B\, z_{T-k}$ with any nonzero $B \in \mathbb{R}^{c \times d}$, then $\hat{\rho}_T(L) = k$.

**Proposition A.2** (Uniqueness of the normalized form for vector outputs). *Fix a matrix norm $\|\cdot\|_{mat}$ on $\mathbb{R}^{c \times d}$. On the domain of nonzero linear maps $L$ of the form* (8)*, there is a unique map $\hat{\rho}_T$ obeying R1-n and R4, namely*

$$\hat{\rho}_T(L) = \frac{\sum_{t=1}^{T} \|B_t\|_{mat}\, \ell(t)}{\sum_{t=1}^{T} \|B_t\|_{mat}}. \tag{10}$$

*Proof of Proposition A.2.* By R4, for linear maps on disjoint time indices the normalized score must be a magnitude-weighted average; with vector outputs the magnitudes are the matrix norms $\|B_t\|_{\mathrm{mat}}$. Thus

$$\hat{\rho}_T(L) = \frac{\sum_{t=1}^{T} \|B_t\|_{\mathrm{mat}}\, \ell(t)}{\sum_{t=1}^{T} \|B_t\|_{\mathrm{mat}}}.$$

On the domain of nonzero maps $L$, we have $\sum_{t=1}^{T} \|B_t\|_{\mathrm{mat}} > 0$, so the denominator is strictly positive. R1-n calibrates single-step maps to $\ell(t)$, which uniquely fixes the form. □

**Takeaway.** On vector-output linear maps, the unnormalized sum of matrix-norm–weighted lags and its normalized average are uniquely determined by minimal, natural rules. This justifies using (3)–(4) with $w_t = \frac{1}{T-t} \sum_{s=t+1}^{T} \|J_{s,t}\|_{\mathrm{mat}}$ as canonical summaries of temporal influence for nonlinear policies via local linearization.

## A.4 ANALYTICAL CALIBRATION OF TEMPORAL RANGE

We compute Temporal Range exactly in two differentiable settings where all Jacobians are available in closed form, now with *vector* outputs and matrix norms.

### A.4.1 EXACT RANGE IN COPY-$k$

Fix a sequence length $T$ and inputs $X_{1:T} = [x_1, \ldots, x_T]$, $x_t \in \mathbb{R}^d$. Consider the map $F : \mathbb{R}^{T \times d} \to \mathbb{R}^c$ that *copies* a linear readout of the observation from $k$ steps ago at the final time:

$$y_T(X) = U\, x_{T-k}, \qquad U \in \mathbb{R}^{c \times d}, \quad k \in \{0, \ldots, T-1\}.$$

In the multi-output formulation, outputs at earlier steps $s < T$ are zero, so $J_{s,t}(X) = 0$ for all $s < T$ and all $t$. The only nonzero Jacobian is The Jacobian block of $y_T$ with respect to $x_t$ is

$$J_{T,t}(X) = \frac{\partial y_T}{\partial x_t}(X) = \begin{cases} U, & t = T-k, \\ 0, & \text{otherwise.} \end{cases}$$

Thus averaging over $s \in \{t+1, \ldots, T\}$ yields $w_t(X) = \frac{1}{T-t} \|U\|_{\mathrm{mat}} \mathbf{1}\{t = T-k\}$ for $t < T$; the normalization factor $(T-t)^{-1}$ cancels in $\hat{\rho}_T$. With the lag $\ell(t) = T - t$, the unnormalized and normalized ranges are

$$\rho_T(F; X) = \sum_{t=1}^{T} w_t\, \ell(t) = \|U\|_{\mathrm{mat}} \cdot k, \qquad \hat{\rho}_T(F; X) = \frac{\sum_t w_t\, \ell(t)}{\sum_t w_t} = k.$$

Thus $\hat{\rho}_T = k$ **exactly**, matching the ground-truth offset.

**Invariances in this setting.** If we apply uniform output rescaling $\tilde{y}_T = \alpha y_T$, then all $w_t$ scale by $|\alpha|$, leaving $\hat{\rho}_T$ unchanged. If we uniformly rescale inputs $x_t^\star = \beta x_t$ with $\beta \neq 0$, then $\|\partial y_T / \partial x_t^\star\|_{\mathrm{mat}} = \|\partial y_T / \partial x_t\|_{\mathrm{mat}} / |\beta|$ for all $t$, again leaving $\hat{\rho}_T$ unchanged (while $\rho_T$ rescales).

### A.4.2 EXACT RANGE UNDER LINEAR RECURRENT READOUT

Consider a linear recurrent "memory"

$$h_{t+1} = Ah_t + Cx_{t+1}, \qquad h_0 = 0,$$

and define the vector output at the final time by a linear readout $y_T = Q\,h_T$ with $Q \in \mathbb{R}^{c \times p}$. (For simplicity we present the single final-output case; if outputs are emitted at each step $y_s = Qh_s$, then $w_t$ becomes the average of $\|QA^{s-t-1}C\|_{\mathrm{mat}}$ over $s \in \{t+1, \ldots, T\}$, exhibiting similar exponential decay.) Unrolling,

$$h_T = \sum_{t=1}^{T} A^{T-t}C\,x_t \quad \Longrightarrow \quad \frac{\partial y_T}{\partial x_t} = Q\,A^{T-t}C \in \mathbb{R}^{c \times d}.$$

For any matrix norm $\|\cdot\|_{\mathrm{mat}}$ on $\mathbb{R}^{c \times d}$, the per-step weight is

$$w_t = \left\|Q\,A^{T-t}C\right\|_{\mathrm{mat}} = \left\|Q\,A^{\ell(t)}C\right\|_{\mathrm{mat}}.$$

Hence the influence profile $t \mapsto w_t$ is governed by powers of $A$.

**Takeaway.** In a linear recurrent readout, $w_t$ is determined by the propagator powers $A^\ell$, and $\hat{\rho}_T$ becomes the expected lag under those induced weights. This bridges spectral properties of the memory dynamics and the measured temporal range.

### A.4.3 FROM EXACT SETTINGS TO COMPLEX SIMULATORS

The two cases above show that (i) when dependence is concentrated at a single offset (COPY-$k$), the normalized range recovers $k$ exactly; and (ii) when dependence is distributed and exponentially decaying (linear recurrence), $\hat{\rho}_T$ has a closed form that grows with the effective memory timescale. In complex environments, the same definitions apply; we compute Jacobians with respect to *observations* by differentiating the policy. When the policy is not amenable to automatic differentiation or is only available as a black box, we compute the same $w_t$ via reverse mode on a compact LEM proxy (see Section 4); in practice the resulting $\hat{\rho}_T$ tracks ground-truth diagnostics and performance-relevant memory in control tasks.

### A.5 USE OF LARGE LANGUAGE MODELS

We used large language model (LLM) assistants for writing support: reorganizing sentences and paragraphs for clarity, tightening prose, fixing grammar and LaTeX formatting, and suggesting alternative phrasings.

### A.6 COMPARISON OF AGGREGATION OPERATORS

We investigated using $\phi = \max$ instead of $\phi = \mathrm{mean}$ for the influence weight computation. Table 6 shows TR values with max aggregation.

Table 6: Temporal Range $\hat{\rho}_T$ using $\phi = \max$ aggregation (steps; mean $\pm$ std over episodes). Values cluster around 15–18 for all tasks, reducing discriminative power compared to mean aggregation (Table 2).

| Environment | LEM | GRU | LSTM | LinOSS |
|---|---|---|---|---|
| CartPole | $15.97 \pm 0.24$ | $16.12 \pm 0.21$ | $16.37 \pm 0.57$ | $16.07 \pm 0.32$ |
| Stateless CartPole | $15.96 \pm 0.39$ | $16.14 \pm 0.44$ | $17.11 \pm 1.66$ | $15.99 \pm 0.78$ |
| Noisy Stateless CartPole | $15.71 \pm 0.59$ | $15.67 \pm 0.59$ | $17.78 \pm 1.48$ | $15.91 \pm 0.44$ |
| RepeatFirst | $19.52 \pm 1.79$ | $16.88 \pm 1.01$ | $16.72 \pm 1.56$ | $21.85 \pm 0.98$ |
| Copy $k = 1$ | $16.00 \pm 0.22$ | $15.96 \pm 0.25$ | $15.47 \pm 0.31$ | $15.46 \pm 0.29$ |
| Copy $k = 3$ | $15.94 \pm 0.20$ | $15.94 \pm 0.22$ | $18.94 \pm 1.19$ | $15.37 \pm 0.45$ |
| Copy $k = 5$ | $16.70 \pm 0.43$ | $16.03 \pm 0.26$ | $18.82 \pm 1.40$ | $15.77 \pm 0.49$ |
| Copy $k = 10$ | $17.93 \pm 0.71$ | $17.35 \pm 0.88$ | $18.06 \pm 2.62$ | $17.42 \pm 0.83$ |

Using max causes TR values to cluster, reducing discrimination between tasks. For example, Copy $k = 1$ yields TR $\approx$ 15–16, nearly identical to Copy $k = 10$ at $\approx$ 17–18. With mean aggregation (Table 2), Copy $k = 1$ produces TR $\approx$ 8–12 while Copy $k = 10$ produces TR $\approx$ 12–17, correctly reflecting increased memory demands. The max operator is dominated by the single largest Jacobian norm, which tends to be similar across tasks. Mean provides a more robust measure of *sustained* temporal influence.

## A.7 ADDITIONAL TEMPORAL INFLUENCE PROFILES

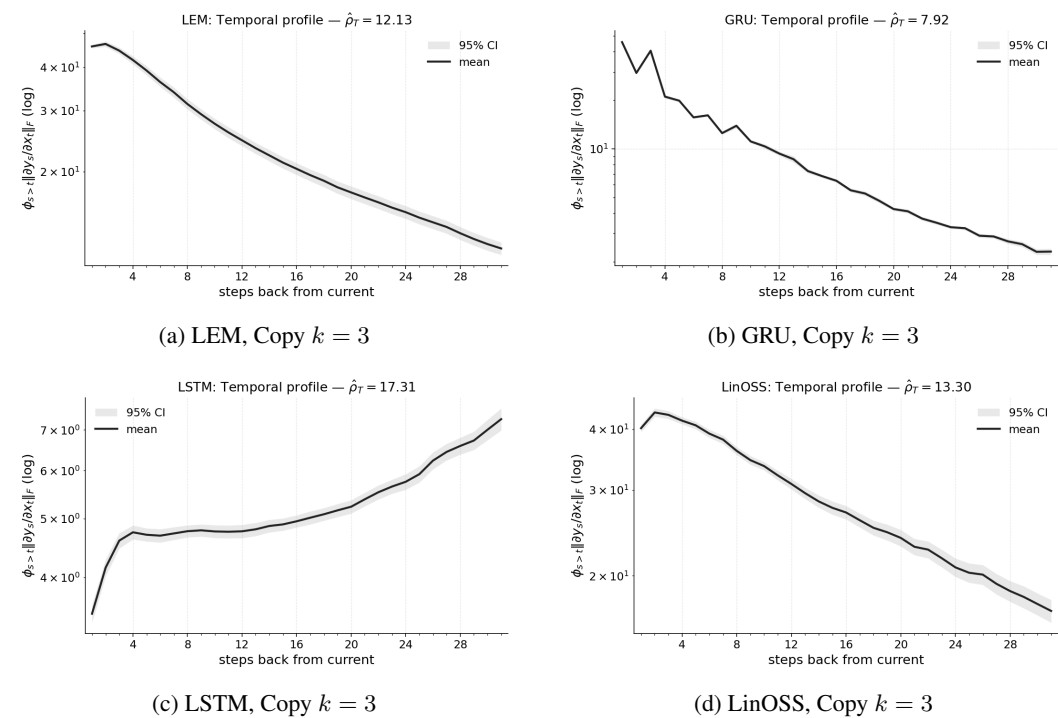

(a) LEM, Copy $k = 3$

(b) GRU, Copy $k = 3$

(c) LSTM, Copy $k = 3$

(d) LinOSS, Copy $k = 3$

Figure 3: Temporal influence profiles for Copy $k = 3$ across all architectures (LEM, GRU, LSTM, LinOSS).

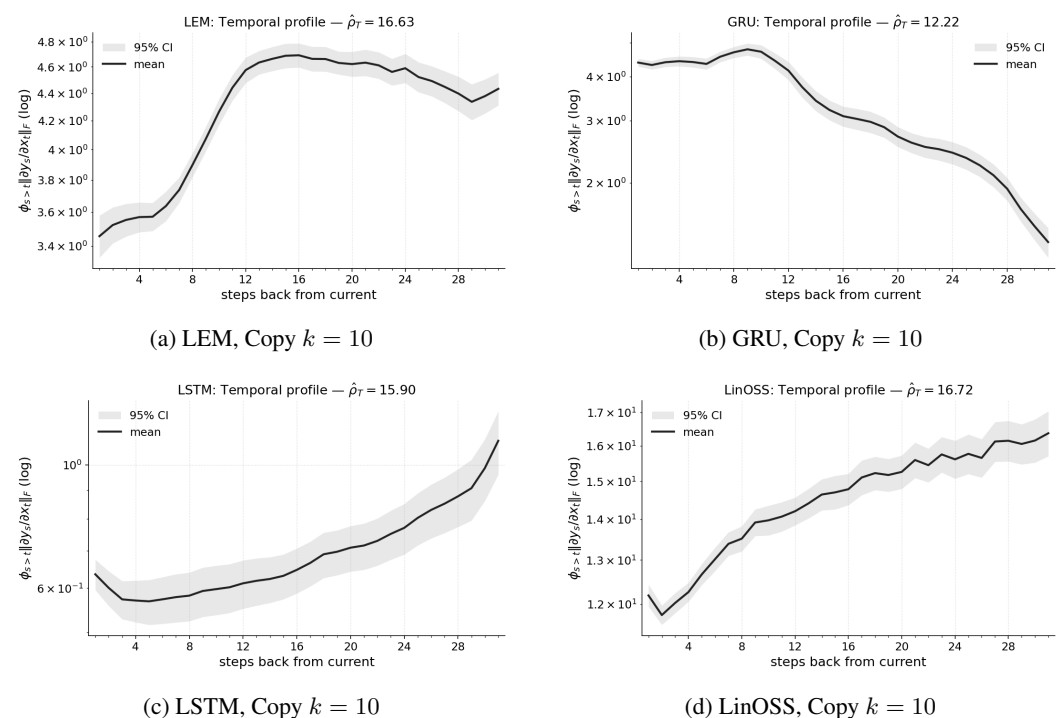

Figure 4: Temporal influence profiles for Copy $k = 10$ across all architectures (LEM, GRU, LSTM, LinOSS).

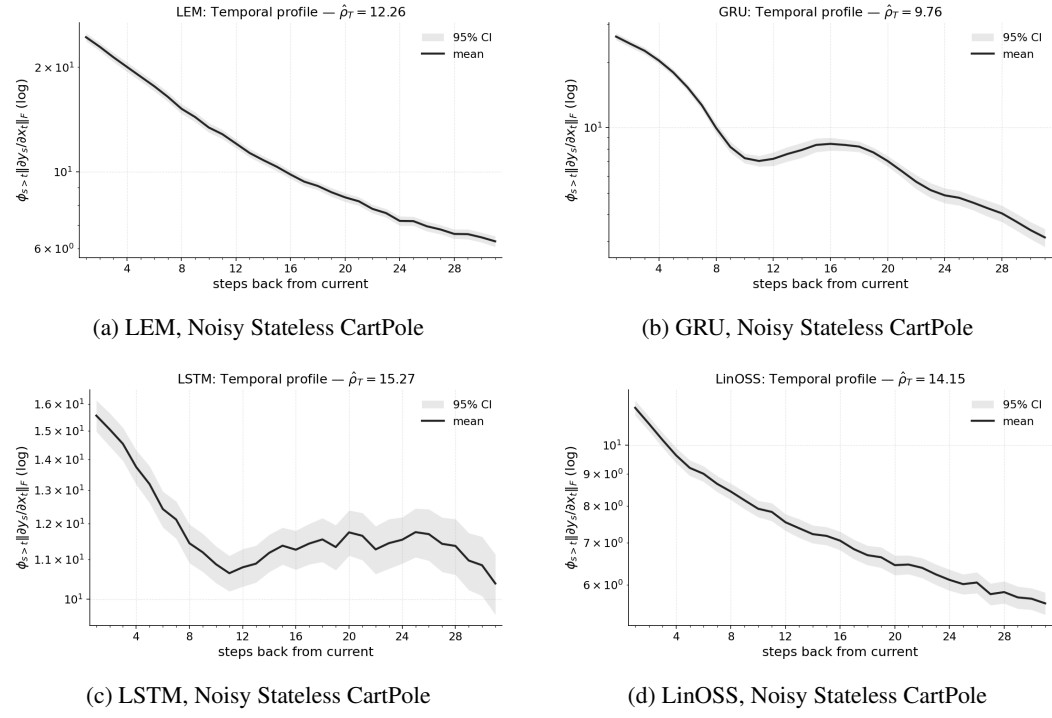

Figure 5: Temporal influence profiles for Noisy Stateless CartPole across all architectures (LEM, GRU, LSTM, LinOSS).

## A.8 HYPERPARAMETERS

See Table 7.

Table 7: Training and evaluation hyperparameters used in all experiments.

| Hyperparameter | Value |
|---|---|
| LR | 3e-4 |
| KL_COEF | 0.0 |
| KL_TARGET | 0.01 |
| NUM_ENVS | 64 |
| NUM_STEPS | 256 |
| TOTAL_TIMESTEPS | 1e7 |
| UPDATE_EPOCHS | 8 |
| NUM_MINIBATCHES | 4 |
| GAMMA | 0.99 |
| GAE_LAMBDA | 0.95 |
| CLIP_EPS | 0.2 |
| ENT_COEF | 0.01 |
| VF_COEF | 0.5 |
| MAX_GRAD_NORM | 0.5 |
| HIDDEN_SIZE | 128 |
| DENSE_SIZE | 128 |
| GRU_HIDDEN_SIZE | 128 |
| LEM_DT | 0.5 |
| ANNEAL_LR | True |
| NUM_TRIALS | 3 |
| T_JACOBIAN | 32 |

