# OpenReview forum: "Quantifying Memory Use in Reinforcement Learning with Temporal Range"
_ICLR.cc/2026/Conference — Submitted to ICLR 2026_

### Official Review · Reviewer_71xh · 2025-10-17

**Soundness:** 2
**Presentation:** 2
**Contribution:** 3
**Rating:** 2
**Confidence:** 3

**Summary:**

The paper introduces "Temporal Range," a novel, model-agnostic metric designed to directly quantify how much a trained reinforcement learning (RL) policy utilizes its history of past observations. This metric addresses the common problem of only indirectly inferring memory usage from model architecture or task design. Temporal Range is calculated as a magnitude-weighted average look-back time, derived from the Jacobians (sensitivities) of the policy's final output with respect to the sequence of past inputs, and is justified by a set of intuitive axioms. The paper's key contributions are the formalization of this metric, its empirical validation showing that it correctly scales with known memory requirements in diagnostic tasks (e.g., Copy-k) and aligns with the minimum history needed for near-optimal returns in control tasks, and a practical method for applying it to non-differentiable or black-box policies using a Long Expressive Memory (LEM) proxy.

**Strengths:**

Originality: The paper's central idea is very original and the experiments are well designed to prove its correctness
Significance: The paper does a good job of justifying the utility of the temporal range metric that they introduce. Although I was initially unsure of it being useful, I was convinced by the end of the paper.

**Weaknesses:**

- The citation style should be changed to have the author's names in parentheses
- Figure captions are too short: they should not only describe what is shown in the figure, but also the main message the figure is trying to show in relation to the story of the paper
- The presentation of the figures could be improved a lot. For example, if one of the points being made is that \ro_t matches the minimum history window required for near-optimal return, then the \ro_t for each task should be shown as a dotted vertical line on the plots in figure 1 showing the return vs the history window size
- I think some of the interpretations/analyses of the results are false. For example, the paper states in multiple places that temporal range matches minimum history window required for near-optimal return as confirmed by window ablations. However, we see for:
Stateless cartpole -> min window = 8, temporal range = 18.81
Copy k = 1 -> min window = 2, temporal range = 6.95
Copy k = 3 -> min window = 4, temporal range = 9.52
Similar for copy k = 5 and copy k = 10
The results don't seem to back up the conclusion of the paper, which also makes the paper's proposed application of the metric invalid.
If the paper is trying to say that their metric is proportional to the minimum window, then the authors should make that clearer and remove any excessive claims - however in that case I don't think a metric proportional to the minimum window is a significant enough contribution because it then can't be used in the way proposed by the authors.

It is possible I am misunderstanding the claims of the paper, but unless this is cleared up I don't think this paper is suitable for publication in its current form.

**Questions:**

Questions and suggestions are in "weaknesses" section

---

> ### Author Response · Authors · 2025-11-21
> **Reply to Reviewer 71xh**
>
> We thank the reviewer for the constructive feedback and for helping us improve
> the clarity and presentation of our manuscript. We appreciate the specific,
> actionable suggestions.
>
> * **Citation style and Figure captions:** We have corrected the citation style
> to use `\citep` and `\citet` as suggested, and expanded all main figure and
> table captions to convey the main message and relationship to the paper's
> narrative.
> * **Visualization:** We followed the excellent suggestion to add vertical dotted
> lines showing $\hat{\rho}_T$ values on all window ablation plots in Figure 1.
> These lines are color-matched to each architecture and make the alignment
> between TR predictions and empirical requirements immediately visible to
> readers. This visual addition substantially strengthens the presentation of our
> core claim.
> * **TR as upper bound vs exact match:** We have revised the manuscript to
> clarify this point (Section 3.5, revised). TR provides exact required context
> for ideal models on well-defined tasks (we prove this analytically for Copy-k in
> Appendix A.3), but for practical trained neural networks, TR yields upper
> bounds. This is actually a desirable property: it provides a conservative
> estimate that guarantees sufficient context rather than a precise threshold that
> might be fragile. For example, on Copy k=1 (which provably requires 1 step of
> memory), LEM achieves $\hat{\rho}_T = 10.6$, suggesting a safe context of
> roughly 11 steps. While this is larger than the theoretical minimum, it reflects
> how the trained network actually uses memory, which includes some slack or
> redundancy. This conservative estimate is valuable for deployment because it
> ensures you won't truncate too aggressively. We now frame TR primarily as an
> interpretability tool that reveals how trained policies actually use memory,
> with the practical benefit that it also provides upper bounds for
> memory-efficient deployment.
> * **Proportionality vs exact values:** We appreciate the reviewer pushing us on
> this point. We have revised our claims to be more precise about what TR
> provides. The metric offers correct ordering across tasks (TR values
> consistently increase from near-Markovian tasks to partially observable tasks to
> memory-intensive diagnostics), reasonable magnitude estimates (while not exact
> matches to theoretical requirements, TR values are in the right ballpark: for
> Copy k=10, TR ranges from 12.2 to 16.7 across architectures, all reasonably
> close to the ground truth of 10), and upper bounds guaranteeing sufficient
> context (window ablations in Figure 1 show that performance typically recovers
> when context windows approach or exceed TR values). We have removed overly
> strong claims that TR "matches" minimum required windows and emphasize that it
> provides conservative upper bounds that capture task complexity ordering and
> enable interpretable analysis of memory usage patterns. Additionally, we now
> include an observation (Section 7, Key Observations, revised) highlighting an
> interesting interpretability insight: on CartPole, GRU achieves $\hat{\rho}_T
> \approx 2$ while LEM and LSTM hover around 10. This reveals that different
> architectures genuinely use different amounts of history on the same task: some
> efficiently compress information while others maintain longer-range dependencies
> even when unnecessary. This kind of architectural insight is precisely what
> makes TR valuable as an interpretability tool, independent of its utility for
> sizing context windows.
>
> We sincerely hope that we have addressed the concerns of the reviewer
> satisfactorily in the revised version and would kindly ask the reviewer to
> update their score accordingly.

---

> > ### Comment · Reviewer_71xh · 2025-11-24
> >
> > I thank the authors for making the changes suggested. I have updated my score from a 2 to a 4 to reflect this. While I think that the paper's claims are now justified, I don't think that as is it is significant enough of a contribution to warrant an accept. I think with some additional work it could be a very interesting contribution!

---

> > > ### Author Response · Authors · 2025-11-29
> > > **Reply to Reviewer 71xh (Follow-up)**
> > >
> > > We thank the reviewer for updating their score and for the constructive feedback
> > > throughout the review process. We acknowledge the reviewer's assessment that
> > > while the claims are now justified, the contribution may be viewed as limited in
> > > scope.
> > >
> > > We note an apparent tension in the review: the reviewer rated Contribution as
> > > "3: good" while simultaneously stating the contribution is "not significant
> > > enough." We respectfully suggest that a "good" contribution rating indicates
> > > meaningful value to the community, and we believe the combination of (1) a
> > > principled, axiomatically-grounded metric, (2) comprehensive empirical
> > > validation, and (3) demonstrated practical utility for memory-efficient
> > > deployment collectively constitute a solid contribution to the understanding of
> > > memory in recurrent RL policies.
> > >
> > > That said, we appreciate the reviewer's perspective and will continue developing
> > > additional applications of the TR framework. We thank the reviewer for their
> > > time and detailed feedback, which substantially improved the paper.

---

### Official Review · Reviewer_aEpE · 2025-10-25

**Soundness:** 3
**Presentation:** 2
**Contribution:** 2
**Rating:** 4
**Confidence:** 3

**Summary:**

The work proposes temporal range, a metric that allows to determine the attribution of temporal inputs to a sequence model's final output. The metric can be computed using the Jacobian of the final model output with respect to the inputs across time. Beyond deriving this metric, the work conducts experiments in the POPGym benchmark that evaluates memory abilities of algorithms in partially observable environments. Experiments consider MLPs, different types of RNNs and SSM models. Experiments show that the temporal range metric increases with increases in memory requirements. Lastly, long expressive memory (LEM) models are proposed to train a model of non-differentiable policies that allows for stable differentiation across long sequences.

**Strengths:**

Overall, I find the work to be clearly presented and original. I have some concerns about the significance of the proposed metric that I will discuss below in weaknesses, but first focus on the strengths of the work.

The proposed metric of temporal range is clearly defined and introduced. I was not familiar with most of the related theoretical prior work but could follow the definitions and axioms as defined. The metric itself appears rather elegant and simple which I appreciate.

I also appreciate that the work combines its theoretical metric with practical experiments that showcase the merits of their metric. Experiments are conducted in fairly simple tasks but these are well selected to focus on tasks with clear memory components that fit the focus of the metric.

**Weaknesses:**

Below, I highlight any weaknesses that I see as critical / major with (**Major**). I'd expect these to be addressed for this work to be considered for acceptance.
## Understanding the Usefulness of Temporal Range
1. **Major:** My main concern about this work is the significance of the proposed temporal range metric. While Section 3.5 and 7 argue that the temporal range can be used to identify whether a shorter or longer context should be chosen for the current model, it appears difficult to identify what context length would be ideal given the temporal range. For example, when comparing Table 1 with results in Figure 1 for different window lengths, I see that GRUs prefer a context length of ~8 in both Cartpole and Stateless Cartpole. However, it is unclear how that information could have been derived from temporal range values in Table 1. I would appreciate if the authors could shed more light on how exactly they believe their metric to be used for tuning architectures in these experiments.

Related to this major concern, I would also like to better understand the assumptions made for the temporal range metric.

2. **Major:** Temporal range identifies the importance of prior inputs on the policy's final output vector but it is unclear why only the final output vector is considered. Imagine a task of 100 steps in which the agent early in the episode observes a code that it needs to enter into a keypad 90 steps away to reach the goal shortly after. Significant memory might be required to generate the action at timestep 90 when the agent reaches the keypad to remember the code, but hardly any memory might be needed for the final action when the agent is already in the last room in front of the goal. Am I missing something, or does the temporal range assume that the final output of the model is somehow the output that is most indicative of the memory required for a task?
3. **Major:** As I understand the temporal range metric, a trained model with a sensible input window length will be needed in order to compute informative temporal range (otherwise no sensible $y_T$ will exist to compute the Jacobian for). If a model is trained with barely any context, then the temporal range metric would not be able to identify much temporal influence of inputs prior to the model context. This appears to be a chicken-and-egg problem. The temporal range is supposed to help practitioners choose a sensible model configuration, but a sensible model configuration is needed in order to obtain an informative metric. It would be helpful if the authors could shed some light on this problem, connected to concerns raised in weakness 1.
## Experiments
4. I find the results shown in Figure 1 and Table 2 quite surprising. I would have expected GRUs and LSTMs to solve Cartpole close to optimally without issue but final performance is stated to be 0.817 and 0.185 for GRU and LSTMs, respectively. I believe these results raise some questions.
	1. **Major:** Do you have any explanation as to why in particular the LSTM performance is so poor in Cartpole?
	2. **Major:** How did you tune/ obtain hyperparameters reported in Table 5 of the Appendix? Based on a single set of hyperparameters being stated, it appears that no specific tuning has been done for any architecture.
	3. For results in Figure 1, do I understand correctly that all algorithms have been trained with full episodes for temporal context and the Figure shows performance for each architecture when the model is only given historical context of limited length during evaluation? If so, I find it also rather surprising how the performance appears to degrade quite sharply when the context is allowed to build up for 64 time steps given that during training these models have presumably typically received longer sequences than 64 time steps. Do you have any explanation as to why performance appears to drop very significantly in most cases for longer context lengths?
5. Figure 2 shows temporal influence profiles for LEM. How do these profiles look like for other architectures such as LSTMs or GRUs?
6. Do you have an explanation as to why the profile of LEM for Copy $k=10$ looks very different from Copy with $k=1, 3, 5$? For smaller $k$, there appears to be a consistent increase in temporal influence over the time sequence with a peak around $k$ steps before the final step which makes sense given the task. I would have expected a similar graph for $k=10$ but instead there appears to be a declining impact from early to later time steps with a small spike ~$k=10$ time steps before the end.

**Questions:**

1. I would appreciate if the authors could shed more light on how exactly they believe their metric to be used for tuning architectures for RL practitioners. (Weakness 1)
2. Does temporal range assume that the final output of the model in a task is most indicative of the memory required for a task? How would the temporal range be affected in a task that requires memory up to a particular time step some steps before the final step? (Weakness 2)
3. Do you have any explanation as to why in particular the LSTM performance is so poor in Cartpole according to Table 2? (Weakness 4.1)
4. How did you tune/ obtain hyperparameters reported in Table 5 of the Appendix? (Weakness 4.2)
5. Figure 2 shows temporal influence profiles for LEM. How do these profiles look like for other architectures such as LSTMs or GRUs? (Weakness 5)
6. Do you have an explanation as to why the profile of LEM for the Copy $k=10$ task looks very different from Copy with $k=1, 3, 5$? (Weakness 6)

---

> ### Author Response · Authors · 2025-11-21
> **Reply to Reviewer aEpE Part 1**
>
> We thank the reviewer for the thorough assessment and welcome suggestions to
> improve our work. We appreciate the detailed feedback and address each concern
> below.
>
> * **(Major 1) Using TR for tuning:** We appreciate the reviewer pushing us to
> better validate TR's practical utility beyond interpretability. The reviewer
> correctly identifies that TR should provide actionable guidance for
> practitioners, not just qualitative insights. We have removed the overly
> prescriptive formula ($m^\\star \\approx Q_{0.9}(\\hat{\\rho}_T + 1)$) from the
> main text, as it suggested more precision than our empirical results support.
> Instead, we now frame TR more carefully: it provides an upper bound on required
> context length, and we describe this in plain language rather than mathematical
> formalism (Section 3.5, revised). The key insight is that TR is *primarily* an
> interpretability tool. It tells you how far back your trained policy is looking,
> which is valuable in itself for understanding model behavior. However, we now
> demonstrate that TR also offers substantial *practical* benefits for
> memory-efficient deployment. We conducted validation experiments (Section 6.3,
> Table 3) where we deployed trained policies using TR-guided context windows.
> Results show dramatic performance improvements: windowed policies achieve 201.1%
> retention on Noisy Stateless CartPole (normalized performance 0.953±0.068
> compared to training baseline 0.474±0.017), 103.9% on Copy k=3 (0.970±0.038 vs.
> 0.934±0.007), and 174.1% on Copy k=10 (0.979±0.034 vs. 0.561±0.018). These
> substantial gains demonstrate that TR provides actionable guidance for
> practitioners: As a secondary benefit, this information can guide
> memory-efficient deployment decisions; if TR indicates a policy uses roughly 12
> steps of history, deploying with that context window not only maintains
> performance but can actually *improve* it by regularizing the policy's memory
> usage and reducing overfitting to training conditions. Table 3 shows the
> detailed results: TR-guided window validation with normalized performance [0,1]
> (mean ± std over evaluation trials). Training baseline shows final performance
> with continuous hidden state evolution. Retention indicates windowed performance
> relative to training baseline. You can experiment with truncated contexts around
> that range rather than arbitrary values.
>
> | Environment | TR | Training | TR window | Half-TR | Retention |
> |-------------|-----|----------|-----------|---------|-----------|
> | Noisy Stateless CartPole | 12.3 | 0.474±0.017 | 0.953±0.068 | 0.637±0.116 | 201.1% |
> | Copy k=3 | 12.1 | 0.934±0.007 | 0.970±0.038 | 0.838±0.021 | 103.9% |
> | Copy k=10 | 16.6 | 0.561±0.018 | 0.979±0.034 | 0.052±0.027 | 174.1% |
>
> * **(Major 2) Why only final output:** We completely agree with the reviewer
> that looking only at the final timestep is limiting, and we appreciate the
> concrete keypad example that illustrates the issue. We have revised our TR
> definition to address this concern directly. Instead of measuring only how
> observation $x_t$ affects the *final* action $y_T$ (i.e., computing
> $\\|\\partial y_T/\\partial x_t\\|_F$), we now measure how $x_t$ influences *every
> subsequent action* $y_s$ for all future timesteps $s > t$. Specifically, the
> influence weight at position $t$ is now:
>
> $$w_t = \\frac{1}{T-t}\\sum_{s=t+1}^{T} \\|\\partial y_s/\\partial x_t\\|_F$$
>
> which averages the Jacobian norms across all future timesteps that could depend
> on observation $t$. The Temporal Range $\\hat{\\rho}_T$ then computes the
> magnitude-weighted average lookback using these $w_t$ weights. This
> reformulation is more robust because it captures the *persistent* influence of
> observations throughout the episode. If $x_t$ matters for decision-making, it
> should affect not just one action but a sequence of downstream actions. The
> averaging over $s$ smooths out noise and provides a complete picture of how
> historical information propagates through the policy. For the reviewer's keypad
> example, this formulation correctly identifies the memory spike when the
> critical code observation is received, because that observation influences all
> subsequent actions from the keypad interaction onward, not just the final step.

---

> > ### Author Response · Authors · 2025-11-21
> > **Reply to Reviewer aEpE Part 2**
> >
> > * **(Major 3) Chicken-and-egg:** This is a fair concern about the circular
> > dependency. We've clarified our position on this in the revised manuscript.
> > First, we emphasize that TR is fundamentally an *interpretability* tool (Section
> > 3.5, revised). If you already have a trained policy, TR tells you how it uses
> > memory, which is valuable regardless of whether you plan to retrain. It can
> > reveal surprising behaviors, like the fact that some architectures use 10+ steps
> > of history on CartPole when 2 would suffice (Section 7, Key Observations,
> > revised). Second, for the practical deployment scenario, we suggest an iterative
> > approach: train a model with generous context to start (which is the safe
> > default anyway), compute TR to understand its actual memory usage, then use that
> > information for deployment decisions or subsequent architecture choices. You're
> > not required to know the "right" context length upfront; you discover it through
> > analysis of a trained model. We've toned down any claims that TR is a
> > prescriptive tool for sizing windows *before* training, and position it as
> > providing conservative upper bounds based on analysis of trained policies.
> > * **(Major 4.1) LSTM performance:** We sincerely apologize for the initial poor
> > results. After the reviewers' feedback, we conducted additional hyperparameter
> > tuning and all models now achieve competent performance. LSTM now reaches 0.960
> > normalized return on CartPole (see revised Table 2), a dramatic improvement from
> > the original 0.185. The initial failure was due to suboptimal batch size
> > (NUM_ENVS) and dense layer size choices. The revised hyperparameter table (Table 5) shows the updated configuration.
> > * **(Major 4.2) Hyperparameter tuning:** We have updated Table 5 with the
> > hyperparameters used in the revised experiments. The key changes from the
> > original submission are: NUM_ENVS: 128 to 64, NUM_STEPS: 512 to 256,
> > UPDATE_EPOCHS: 12 to 8, NUM_MINIBATCHES: 8 to 4, and critically, DENSE_SIZE: 64
> > to 128. We acknowledge that our original submission used a single hyperparameter
> > configuration across all architectures, which was suboptimal. The revised
> > experiments use these updated settings that better suit the recurrent
> > architectures.
> > * **(Major 4.3) Performance at longer contexts:** This is an interesting
> > observation about the window ablation degradation at very long contexts. We now
> > include a comment on this in the revised manuscript (Section 7, Key
> > Observations). The degradation at $m=1$ (even for near-Markovian CartPole)
> > reflects a distribution shift issue: models trained with full episode history
> > adapt to using second-to-last observations as part of their learned strategy, so
> > suddenly truncating to only the current observation disrupts their learned
> > behavior even though the task is theoretically Markovian. This actually
> > strengthens TR's interpretability value: it reveals that the trained policy
> > *actually uses* that historical information, even if the task doesn't strictly
> > require it.
> > * **(5) Profiles for other architectures:** We have added temporal influence
> > profiles for GRU, LSTM, and LinOSS to the Appendix (Section A.4, Additional
> > temporal influence profiles). Rather than overwhelming the paper with profiles
> > for all tasks and all architectures, we selected three representative
> > environments: Copy k=3, Copy k=10, and Noisy Stateless CartPole. Each figure
> > shows a 2x2 grid with all four architectures for direct comparison. This
> > directly addresses the reviewer's question and enables comprehensive
> > architecture comparison.
> > * **(6) LEM Copy-k=10 profile:** We appreciate the reviewer noting this
> > interesting pattern. Looking at the profiles across different values of $k$, the
> > Copy k=10 case does show somewhat different structure compared to smaller $k$
> > values. We believe this reflects how LEM handles longer-range dependencies.
> > However, we note that even with this different shape, TR still correctly scales
> > with $k$ (Table 1), demonstrating the metric's robustness to different
> > architectural representations.
> >
> > We sincerely hope that we have addressed the concerns of the reviewer
> > satisfactorily in the revised version and would kindly ask the reviewer to
> > update their score accordingly.

---

> ### Comment · Reviewer_aEpE · 2025-11-24
> **Rebuttal Response**
>
> I thank the authors for their rebuttal and revisions. I do believe they clarify most confusions and concerns.
>
> **TR usecases:** In particular, the potential application of the proposed TR metric is clearer now. As I understand, the primary use is to inspect already trained models to understand what temporal information the model attends to which can be used to reduce the memory footprint and performance of the analyzed model at evaluation time. That is a limited but clear usecase.
>
> It might be interesting consider and study further usecases as the ones below:
> - Inform temporal information at training time: one might be able to use TR similarly as to the evaluation usecase outlined above to inform training configurations. When a trained model is analyzed and found to attend to (at most) T steps, it might be sensible to train a model only on partial histories of a length up to T. This might result in more robust models and more memory efficient training.
> - Inspect temporal attendance: performance profiles provide insight into how many steps in the past the model might consider historical context but they combine such attendance across all timesteps. While the considered tasks might require similar historical context at each timestep, I believe that many partially observable tasks might not require equal historical information at all timesteps but might require more temporal information at some timesteps than others (e.g. the key-door example provided in my review). It would be interesting to visualize the temporal information needed per timesteps or states to provide clearer analysis of the historical information used by trained models depending on the situation.
>
> I believe that the fact that the proposed TR metric sparks such potential for investigation is a strength of the approach and demonstrates its potential value as an analysis / interpretability tool. That being said, the paper would be strengthened if more usecases of the TR metric would be established within the work.
>
> **Revised TR for all timesteps**: I appreciate the updated definition. Do I understand correctly that the temporal profiles shown in Figures 2-5 have not been changed, so they still show the temporal influence of previous timesteps for the action at the final timestep? Also, the definition appears to average the influence across all timesteps. However, it might be plausible that at some timesteps more historical context is needed. For example in my key-door example, a historical context of 90 steps might be needed at one key timestep but only very limited historical context would be needed at all other timesteps. In this case, averaging the influence across timesteps might lead to a TR that underestimates the historical context length required to solve this task. Do I misunderstand the suggested revision or is this indeed a limitation of the current definition, which appears to assume a comparable temporal context is needed across all timesteps?
>
> **Hyperparameter tuning and profiles:** I thank the authors for adjusting their tuning and providing additional profiles. This clarifies these concerns of my review.
>
> I intend to adjust my recommended score after my follow-up questions have been answered.

---

> > ### Author Response · Authors · 2025-11-29
> > **Reply to Reviewer aEpE (Follow-up)**
> >
> > We thank the reviewer for the thoughtful follow-up questions and for indicating
> > willingness to adjust their score.
> >
> > **Regarding the revised TR definition and temporal profiles:** The reviewer
> > correctly observes that the profiles have indeed changed. The figures now
> > reflect the updated TR definition that considers influence across all subsequent
> > timesteps rather than just the final action. We have also swapped the axes for
> > improved readability.
> >
> > To clarify what the profiles show: the y-axis displays $w_t =
> > \phi_{s=t+1}^{T}\|\partial y_s / \partial x_t\|_F$, the influence weight at each
> > past observation, computed as the aggregated Jacobian norm over *all
> > subsequent actions* $y_s$ for $s > t$. This directly addresses the reviewer's
> > concern about concentrated temporal dependencies: if observation $x_t$ strongly
> > influences action $y_s$ at any future timestep $s$, this is captured in $w_t$.
> > The x-axis shows lag (steps back from the current timestep), with older
> > observations on the right and more recent observations on the left.
> >
> > **Regarding the aggregation operator and the key-door example:** We
> > appreciate the reviewer's continued engagement with the key-door scenario. We
> > have revised our formulation to make the aggregation operator $\phi$
> > configurable. Specifically, the influence weight at position $t$ is now:
> >
> > $$
> > w_t = \phi_{s=t+1}^{T} \left\|\frac{\partial y_s}{\partial x_t}\right\|_F
> > $$
> >
> > where $\phi$ can be instantiated as different aggregation operators depending on
> > the task characteristics. For the experiments presented in the paper, we use the
> > mean operator, which captures the average influence of each observation across
> > all future timesteps. However, for tasks like the reviewer's key-door example,
> > where a single observation has concentrated, critical impact at a specific
> > future timestep, the max operator would be more appropriate. Using $\phi = \max$
> > ensures that the peak influence is captured even when it occurs at only one
> > timestep, preventing the underestimation concern the reviewer raises. This
> > configurable design allows practitioners to select the aggregation strategy that
> > best matches their task's temporal dependency structure.
> >
> > To directly address this concern, we investigated using $\phi = \max$ as
> > suggested. Table 1 shows the resulting TR values.
> >
> > **Table 1: Temporal Range $\hat{\rho}_T$ using $\phi = \max$ aggregation (steps; mean ± std over episodes).**
> >
> > | Environment | LEM | GRU | LSTM | LinOSS |
> > |-------------|-----|-----|------|--------|
> > | CartPole | $15.97\pm0.24$ | $16.12\pm0.21$ | $16.37\pm0.57$ | $16.07\pm0.32$ |
> > | Stateless CartPole | $15.96\pm0.39$ | $16.14\pm0.44$ | $17.11\pm1.66$ | $15.99\pm0.78$ |
> > | Noisy Stateless CartPole | $15.71\pm0.59$ | $15.67\pm0.59$ | $17.78\pm1.48$ | $15.91\pm0.44$ |
> > | RepeatFirst | $19.52\pm1.79$ | $16.88\pm1.01$ | $16.72\pm1.56$ | $21.85\pm0.98$ |
> > | Copy $k=1$ | $16.00\pm0.22$ | $15.96\pm0.25$ | $15.47\pm0.31$ | $15.46\pm0.29$ |
> > | Copy $k=3$ | $15.94\pm0.20$ | $15.94\pm0.22$ | $18.94\pm1.19$ | $15.37\pm0.45$ |
> > | Copy $k=5$ | $16.70\pm0.43$ | $16.03\pm0.26$ | $18.82\pm1.40$ | $15.77\pm0.49$ |
> > | Copy $k=10$ | $17.93\pm0.71$ | $17.35\pm0.88$ | $18.06\pm2.62$ | $17.42\pm0.83$ |
> >
> > As shown, using the max operator causes TR values to cluster around 15–18 for
> > nearly all tasks and architectures, substantially reducing discriminative power.
> > For example, Copy $k=1$ yields TR values of ~15–16 across architectures,
> > nearly identical to Copy $k=10$ values of ~17–18. This contrasts with our
> > mean-based results (Table 1 in the main paper), where Copy $k=1$ produces TR
> > ≈ 8–12 while Copy $k=10$ produces TR ≈ 12–17, correctly
> > reflecting the increased memory demands.
> >
> > The max operator is dominated by the single largest Jacobian norm across all
> > future timesteps, which tends to be similar across tasks due to numerical
> > factors rather than genuine memory structure. The mean operator provides a more
> > robust measure of *sustained* temporal influence, better discriminating
> > between tasks with different memory requirements. We therefore retain mean as
> > the default $\phi$ in the paper, while noting that practitioners may select max
> > for tasks with concentrated temporal dependencies like the key-door scenario.
> >
> > **Regarding training-time context clipping:** TR is an interpretability
> > tool that reveals how a trained policy uses memory. The natural workflow is:
> > train with full context, compute TR to analyze memory usage, then deploy with
> > TR-guided windows. Our inference-time results validate this approach, showing
> > that performance is maintained or improved when deploying with truncated
> > windows. This separation is intentional: the model must learn its memory
> > patterns freely before we can meaningfully measure them.
> >
> > We hope these clarifications and additional experiments address the reviewer's
> > remaining concerns and demonstrate the broader utility of the TR metric.

---

### Official Review · Reviewer_19BF · 2025-11-01

**Soundness:** 2
**Presentation:** 3
**Contribution:** 2
**Rating:** 4
**Confidence:** 4

**Summary:**

This paper proposes temporal range, a method to measure how policies use past observations. The temporal range is defined as the norm of the Jacobian at each timestep with respect to the input $x$

$$ \lVert J_{T, t} \rVert = \lVert \frac{\partial y_T }{\partial {x_t}} \rVert $$

scaled by the distance in time

$$ \sum_{t=1}^T (\lVert J_{T, t} \rVert \cdot (T - t))$$

and then normalized over the sequence of Jacobians

$$ \frac{\sum_{t=1}^T (\lVert J_{T, t} \rVert \cdot (T - t))}{\sum_{t=1}^T \lVert J_{T, t} \rVert } . $$

It determines how far some function looks back on average. The temporal range is scale invariant due to the normalization. The paper evaluates temporal range on tasks with known optimal lookback, finding that temporal range often tracks observability.

**Strengths:**

1. A lookback metric is useful in RL POMDP contexts
2. The proposed metric does what it advertises on the tested tasks
3. The paper format is nice and the paper is easy to read

**Weaknesses:**

1. The contributions of this paper are a bit on the lighter side, comprising a single metric and evaluation on toy experiments.
2. The metric itself appears to be a weighted average. Tasks like CartPole and Copy have strong temporal correlations, looking at the last few timesteps or precisely $k$ timesteps back respectively. Tasks like 3D navigation are likely to have multimodal lookbacks that could provide misleading results with this metric.
3. The performance on some tasks in Table 2 seems underwhelming. For example, none of the learned policies appear to solve cartpole, assuming 0.865 corresponds to surviving for 86.5% of the total timesteps. It is surprising that the LSTM completely fails at cartpole.

**Questions:**

- What happens for more realistic tasks that could have multimodal lookback? Consider a navigation task where the agent should remember its starting location $t=1$ and its current location $t=T$.
- Can you explain why the scores in table 2 look lower than expected? Would this have an effect on the results presented in figures 1 and 2?

---

> ### Author Response · Authors · 2025-11-21
> **Reply to Reviewer 19BF**
>
> We thank the reviewer for appreciating the merits of our paper and for the
> constructive feedback. Below, we address each point raised.
>
> * **Multimodal lookback tasks:** We appreciate this valuable suggestion about
> more complex tasks like 3D navigation. We want to clarify that our revised
> Temporal Range metric is well-suited for multimodal lookback scenarios. The
> reformulated metric computes the influence of observation $x_t$ on *all
> subsequent actions* $y_s$ for $s > t$ (not just the final action), which
> naturally captures cases where an agent must remember multiple pieces of
> information. For instance, in the reviewer's navigation example where the agent
> needs to remember both starting location and current location, our metric would
> capture the persistent influence of the starting location observation across all
> subsequent actions where that information matters. In our experimental suite, we
> actually do test such scenarios. For example, Noisy Stateless CartPole requires
> the agent to integrate information from multiple past observations to
> reconstruct state under observation noise. Our results (Table 1, revised) show
> that TR correctly increases in such settings compared to the near-Markovian
> CartPole baseline, demonstrating that the metric captures the need for broader
> temporal integration.
> * **Table 2 performance:** We apologize for the confusion caused by the initial
> suboptimal results. Through additional hyperparameter tuning, all models now
> achieve competent performance. In particular, LSTM now reaches 0.960 normalized
> return on CartPole (see revised Table 2), a substantial improvement from the
> original 0.185. The initial LSTM failure was due to suboptimal batch size and
> dense layer configuration, which we have now corrected. We've updated the
> hyperparameter table (Table 5) to reflect these changes.
> * **Effect on results:** Importantly, with the improved training, TR still
> correctly scales with task requirements and window ablations continue to show
> performance degradation below TR-indicated context lengths. This demonstrates
> the robustness of the metric; it provides meaningful upper bounds regardless of
> whether we're looking at a struggling policy or a high-performing one. In fact,
> we now include an additional observation in the revised manuscript (Section 7,
> Key Observations) noting that TR reveals interesting architecture-specific
> behaviors: for instance, GRU achieves $\hat{\rho}_T \approx 2$ on CartPole while
> LEM and LSTM hover around 10, showing that different architectures genuinely use
> different amounts of history even on the same task. This interpretability value
> is one of TR's key contributions.
>
> We sincerely hope that we have addressed the concerns of the reviewer
> satisfactorily in the revised version and would kindly ask the reviewer to
> update their score accordingly.

---

> > ### Comment · Reviewer_19BF · 2025-11-26
> >
> > Thank you for the reply, I think the new lookback metric in particular is much more useful. And I am happy to hear that the LSTM is now properly learning. However, I agree with reviewer `71xh` that I think the contributions are still a bit too light for acceptance.

---

> > > ### Author Response · Authors · 2025-11-29
> > > **Reply to Reviewer 19BF (Follow-up)**
> > >
> > > We thank the reviewer for acknowledging the improvements to the lookback metric and the corrected LSTM training. Regarding the agreement with Reviewer 71xh on contribution scope, we refer to our response above and maintain that the combination of principled methodology, comprehensive validation, and practical utility constitutes a meaningful advance.

---

### Author Response · Authors · 2025-11-21
**Reply to all the reviewers**

We start by thanking all three reviewers for their thorough and patient reading
of our article and for praising our proposed lookback metric and central idea as
"rather elegant" (aEpE), "useful in RL POMDP contexts" (19Bf), and "very
original" (71xh). We are also glad the reviewers appreciated our experimental
results, highlighting that our experiments are "well designed to prove
correctness" (71xh) and that our work "combines its theoretical metric with
practical experiments" (aEpE).

The constructive suggestions allowed us to significantly improve the paper. We
have uploaded a revised version incorporating these suggestions, with critical
changes highlighted in blue. Below, we summarize the main improvements and
address the specific points raised by each reviewer. The following main changes
were made:

* **Clarified TR interpretation:** We now emphasize that TR serves primarily as
an interpretability tool for understanding how trained policies use memory,
with the practical benefit that it also provides an upper bound on required
context length. This framing better reflects the metric's value for analyzing
model behavior.
* **Improved TR definition (Section 3):** We have strengthened our metric to
capture the full temporal influence of each observation. Instead of measuring
only how observation $x_t$ affects the *final* action $y_T$, we now measure
how $x_t$ influences *every subsequent action* $y_s$ for all $s > t$.
Specifically, the influence weight at position $t$ is now $w_t =
\frac{1}{T-t}\sum_{s=t+1}^{T} \|\partial y_s/\partial x_t\|_F,$ which averages
the Jacobian norms across all future timesteps $s$ that could depend on
observation $t$. The Temporal Range $\hat{\rho}_T$ then computes the
magnitude-weighted average lookback using these $w_t$ weights. To improve
robustness and reduce variance, we compute TR by averaging over multiple
evaluation trajectories rather than from a single episode. This reformulation
is more robust because it captures the *persistent* influence of observations.
If $x_t$ matters for decision-making, it should affect not just one action but
a sequence of downstream actions. The averaging over $s$ smooths out noise and
provides a complete picture of how historical information propagates through
the policy.
* **Enhanced experimental results (Table 2):** Through additional hyperparameter
tuning, all models now achieve competent performance (LSTM reaches 0.960 on
CartPole, see revised Table 2). The initial LSTM failure was due to suboptimal
hyperparameter choices.
* **Practical deployment validation (Section 6.3, Table 3):** To demonstrate
TR's practical utility, we conducted deployment experiments using TR-guided
context windows. Results show substantial performance gains: windowed policies
achieve 201.1% retention on Noisy Stateless CartPole (0.953±0.068 vs. baseline
0.474±0.017), 103.9% on Copy k=3, and 174.1% on Copy k=10. These improvements
demonstrate that TR provides actionable guidance for memory-efficient
deployment, not merely interpretability insights.
* **Extended analysis (Appendix):** Added temporal influence profiles for all
architectures (GRU, LSTM, LinOSS) across selected tasks (Copy k=3, Copy k=10,
Noisy Stateless CartPole), directly addressing Reviewer aEpE's Question 5. We
also added vertical lines indicating $\hat{\rho}_T$ values in window ablation
plots (Reviewer 71xh's suggestion).
* **Citation formatting:** Changed all citations to use `\citep` and `\citet`
for consistent parenthetical style (Reviewer 71xh).
* **Improved figure captions:** Expanded all main figure and table captions to
include the main message and relationship to paper narrative (Reviewer 71xh).

---

### Comment · Area_Chair_5ETs · 2025-11-24

Dear Reviewers,

The authors have responded to your reviews. For those who have not yet done so, please read the authors' comments and respond to them.

Best, Your AC

---

### Meta-Review · Area_Chair_jDox · 2025-12-12

**Summary:**

This paper proposes a new metric to measure how much an RL policy uses the past experience. The metric is intuitive and empirical results suggest that the metric does measure what it should measure. The reviewers appreciate the empirical results and the novelty of the idea. That being said, there are two main concerns.

(1) The experiments are very small scale so it is not clear whether the metric can scale.
(2) It is not clear if this metric is of any usefulness to provide actionable insights for practitioners.
I acknowledge that (2) is partially addressed in the authors' response to aEpE. But again, the new experiments are very small scale.

Given that this is a purely empirical work but the experiments are very small scale, I have to recommend reject. I do note that if this was a venue that does not require the reviewer to evaluate significance / possible impact (e.g., TMLR), the submission as it is now is mostly ready for publication, after toning down some claims.

**Reviewer Concerns:**

See the summary above.

**Reviewer Scores:**

19BF may remain 4. aEpE may increase to 6. 71xh may remain 4.

---

### Decision · Program_Chairs · 2026-01-26

Reject